# Insights into Non-Exercise Physical Activity on Control of Body Mass: A Review with Practical Recommendations

**DOI:** 10.3390/jfmk8020044

**Published:** 2023-04-11

**Authors:** Diego A. Bonilla, Javier O. Peralta-Alzate, Jhonny A. Bonilla-Henao, Roberto Cannataro, Luis A. Cardozo, Salvador Vargas-Molina, Jeffrey R. Stout, Richard B. Kreider, Jorge L. Petro

**Affiliations:** 1Research Division, Dynamical Business & Science Society—DBSS International SAS, Bogota 110311, Colombia; 2Research Group in Physical Activity, Sports and Health Sciences (GICAFS), Universidad de Córdoba, Monteria 230002, Colombia; 3Research Group in Biochemistry and Molecular Biology, Faculty of Sciences and Education, Universidad Distrital Francisco José de Caldas, Bogota 110311, Colombia; 4Programa Talentos Colombia, Ministerio del Deporte, Apartado 057840, Colombia; 5Semillero de Investigación en Ciencias de la Actividad Física y el Entrenamiento Deportivo (SISCAFED), Complejo Tecnológico, Agroindustrial, Pecuario y Turístico SENA, Apartado 057841, Colombia; 6Galascreen Laboratories, Department of Pharmacy Health and Nutritional Sciences, University of Calabria, Via Savinio, Edificio Polifunzionale, 87036 Rende, Italy; 7Research and Measurement Group in Sports Training (IMED), Faculty of Health Sciences and Sports, Fundación Universitaria del Área Andina, Bogota 111221, Colombia; 8Faculty of Sport Sciences, EADE-University of Wales Trinity Saint David, 29018 Malaga, Spain; 9Physiology of Work and Exercise Response (POWER) Laboratory, Institute of Exercise Physiology and Rehabilitation Science, University of Central Florida, Orlando, FL 32816, USA; 10Exercise & Sport Nutrition Laboratory, Human Clinical Research Facility, Texas A&M University, College Station, TX 77843, USA

**Keywords:** sedentary behavior, physical inactivity, non-communicable diseases, activity trackers

## Abstract

Non-exercise physical activity (NEPA), also called unstructured or informal physical activity, refers to those daily activities that require movement of the human body without planning or strict control of the physical effort made. Due to new technologies and motorized transportation devices, the general population has significantly decreased its NEPA. This increase in sedentary lifestyles, physical inactivity, and excessive energy intake is considered a risk factor for obesity, non-communicable diseases (NCDs), and all-cause mortality. Searching in PubMed/MEDLINE and Web of Science databases, a narrative review of NEPA was carried out to address its conceptualization, promotion strategies for the general population, and monitoring through wearable devices. It is strongly recommended that governmental entities, health practitioners, and the construction industry adhere to “The Global Action Plan on Physical Activity 2018–2030: More Active People for a Healthier World” and implement different salutogenic urban strategies. These strategies aim to generate environments that motivate increases in NEPA, such as cycling and walking transportation (between 5000–12,500 steps per day), and the progression to physical exercise. There is a wide variety of electronic devices for personal use, such as accelerometers, smartphone apps, or “smart clothes”, that allow for the monitoring of NEPA, some with a wide range of analysis variables contributing to the estimation of total daily energy expenditure and the promotion of healthy habits. In general, the further promotion and monitoring of NEPA is required as part of a strategy to promote healthy habits sustainable over time for the prevention and control of obesity and NCDs.

## 1. Introduction

Obesity is multifactorial, highlighting factors such as genetics, age, sex, and nutrition, among many others [1]. Physical inactivity is one of the main causes of this condition that leads to metabolic alterations, considering the importance of the locomotor system and physical effort in the communication between tissues and organs (cross-talking) at a systemic level [2]. The combination of excessive energy intake and physical inactivity act synergistically as the main causes of obesity [3]. Over time, adaptation to a positive caloric balance and physical inactivity generates physiological negative feedback mechanisms to an eventual reduction in energy intake, which is common in body fat loss programs. This is mainly due to the decrease in resting energy expenditure and the fluctuation in the serum concentration of orexigenic/anorexigenic hormones [4]. Besides effective fat mass reduction, another challenge is to avoid regaining it after a diet or exercise intervention. Many people who lose fat mass eventually regain most of it in the long term [5,6]. In recent decades, emphasis was placed on strategies that seek to increase non-exercise physical activity (NEPA), such as walking or active transportation [7], to increase physical activity levels and the subsequent levels of non-exercise activity thermogenesis (NEAT). However, in most people, NEAT is the second largest contributor to total caloric expenditure, and it was reported that individuals with obesity, compared to lean individuals, have a lower NEAT [8], which increases the risk of obesity.

It should be considered that physical inactivity results in a decrease in physical fitness [9], which has an impact on the health of the individual because the latter is also a risk factor for all-cause cardiovascular morbidity and mortality [10,11]. Considering the importance of NEPA in the control of excess weight and obesity, the aim of this review was (i) to evaluate the impact of NEAT on obesity management and indicators, and (ii) to address aspects of NEPA related to conceptualization, promotion strategies for the general population, and quantification through portable devices.

## 2. Methods

Taking into account that narrative synthesis often lacks transparency and that synthesis methods are rarely reported [12], previous guidelines on the development of a narrative review outlined by Dixon-Woods et al. [13] and Peters et al. [14] were followed. In brief, the identification, selection, evaluation, and synthesis of the available literature were performed to summarize the body of knowledge related to NEPA to be used for practitioners [15]. The first version of the manuscript was conducted as part of the capstone project required by the ‘Specialization in Physical Activity and Health’ degree program at Universidad de Córdoba (Montería, Colombia). Experts and researchers in the field collaborated remotely on revising the draft before final approval.

### 2.1. Information Sources

The primary sources for the articles were the following online databases: PubMed/MEDLINE, Web of Science, and Google Scholar. 

### 2.2. Search Strategy

The following Boolean algorithm was used: (“NEAT” OR “non-exercise activity thermogenesis” OR “unstructured physical activity” OR “non-exercise physical activity”) AND obesity. In order to perform some search specifications, reserved symbols were taken into account (e.g., quotation marks to track keywords literally). Additionally, a manual search was performed in Google Scholar. 

### 2.3. Findings Presentation

The narrative discussion was aligned with individual articles and interpretations of relevant articles by organizing it into sections: (i) non-exercise physical activity; (ii) energy expenditure from non-exercise physical activity; (iii) promotion strategies; (iv) technological monitoring of non-exercise physical activity.

## 3. Non-Exercise Physical Activity

From a classical perspective, physical activity is defined as “any bodily movement produced by skeletal muscles that results in an expenditure of energy”, a definition that was adopted by the World Health Organization (WHO) [16] and several organizations and researchers [17,18,19,20]. However, for Piggin [21], this definition may be reductionist or simplistic; therefore, a more holistic definition with implications in the educational context, research, and public policies is required. This author states that physical activity “implies that people move, act, and perform within culturally specific spaces and contexts, and are influenced by a unique variety of interests, emotions, ideas, instructions, and relationships”. We consider that this definition complements the classic concept (mechanical view), as it invites practitioners to reflect on aspects related to why, for what purpose, or how a person performs (or does not perform) physical activity.

Physical activity comprises a wide range of categories or types. In particular, NEPA, which is also described as unstructured or informal physical activity, refers to those activities that require movement of the human body and thereby generate energy expenditure (which depends on the activities’ duration and intensity, among other variables) and that are performed daily without planning or strict control of the physical effort made. This includes climbing stairs, dancing, domestic or work activities, walking, cycling to work, playing, gardening, and walking pets, among many other activities [7]. Unlike physical exercise, NEPA does not have systematic programming that is planned, structured, and purposeful physical activity (e.g., improving physical fitness). However, physical activity includes NEPA, exercise, and other types of body movement [22].

The effects of physical activity on health have been consistently evidenced. In this regard, the reduction of cardiovascular risk and the improvement of body composition with physical exercise programs, both strength and endurance, are widely documented [23,24,25,26]. However, NEPA has not received the same attention, and it is important to consider it as a necessary complement when seeking to control excess body fat and different types of pathologies related to physical inactivity. This has a very important connotation, considering that more than half of the world’s population has insufficient levels of physical activity. This represents a public health problem because physical inactivity is one of the main factors causing mortality worldwide (~6%); in addition, it is the main cause of ischemic heart disease (~30%), diabetes (~27%), and breast and colon cancer (~21% and 25%, respectively) [27].

Besides physical inactivity, sedentary behavior increases the risk of non-communicable disease and mortality. At this point, it should be noted that although often used interchangeably, physical inactivity and sedentary behavior are not the same, which is important for risk factor analysis and intervention strategy purposes. Sedentary behavior is any waking behavior characterized by an energy expenditure of 1.5 metabolic equivalents (METs) while sitting, reclining, or lying down (e.g., watching TV, driving). In contrast, physical inactivity is a level of physical activity insufficient to meet current recommendations for substantial health benefits, which, in adults, the WHO states is “150 to 300 min of moderate-intensity physical activity; or at least 75–150 min of vigorous-intensity physical activity; or an equivalent combination of moderate- and vigorous-intensity activity during the week” [28]. Under this perspective, sedentary behavior and physical inactivity are two behavioral risk factors that are closely related, but each exerts and interacts with each other, affecting health. This has been highlighted in observational studies, in which high sedentary behavior (>8 h/day) and low levels of physical activity (2–5 MET-hours/week) were associated with higher mortality. However, with higher levels of physical activity (35.5 MET-hours/week), this mortality is lower. In particular, people with low sedentary behavior (<4 h/day) and high levels of physical activity (35.5 MET-hours/week) show low mortality compared to the aforementioned conditions [29] (Figure 1).

## 4. Energy Expenditure from Non-Exercise Physical Activity

The reduction of body mass, or more specifically, body fat, is a function of a negative caloric balance induced by dietary manipulation and physical activity, which must be in accordance with the needs of each subject and sustained over time [30]. However, controlling the energy balance is difficult due to several factors that are mainly linked to caloric expenditure. 

The traditional model of total daily energy expenditure in humans is composed of the sum of the energy allocated to the maintenance of the basal metabolic rate, the thermic effect of food, and the physical activity energy expenditure. Basal metabolic rate is the minimum amount of energy expended in all homeostatic processes of the body. It represents the basic energy requirements of our body’s organs (e.g., brain, gut, kidneys, heart, liver, muscle, etc.) and comprises the largest proportion of total daily energy expenditure (~60–70%). Basal metabolic rate is assessed after rest and fasting (10–12 h) with the subject awake, in a prone position, and under thermoneutral conditions. It should be noted that the evaluation of resting energy expenditure is more frequent due to practical considerations that are not met when evaluating basal metabolic rate [31]. The thermic effect of food is the energy expenditure associated with the process of the digestion, absorption, and assimilation processes of food; it represents between ~6–12% of total daily energy expenditure and is a relatively stable component. The thermic effect of food is proportional to caloric intake, and the differences between lean and obese subjects are small; furthermore, there is insufficient evidence to demonstrate a relationship between the thermic effect of food and the development of obesity [32]. The physical activity energy expenditure comprises energy expenditure from physical exercise (structured PA) and NEPA; both components vary widely within and between individuals.

It is important to note that both NEAT and NEPA are not interchangeable but represent complementary concepts; NEAT refers to energy expenditure, whereas NEPA describes types of body activity that are not defined as purposeful movements but contribute to NEAT. Specifically, NEAT corresponds to all of the energy expenditure of all activities that are not physical exercise, such as daily chores or work, leisure activities, sitting, standing, walking, singing, dancing, and others [8]. It should be noted that a certain percentage of spontaneous physical activity, which is part of NEAT, is beyond voluntary control (e.g., "restless leg movement" while sitting).

Studies on NEAT have focused their attention on the prevention and control of obesity. In this regard, a study by Levine et al. [33] showed that obese subjects sat 164 min more per day than lean subjects; also in addition, lean people were reported to be upright and walking for 152 min more per day than obese participants. According to the authors, if obese individuals adopted behaviors to increase NEAT (e.g., increasing NEPA by walking) similar to lean subjects, then they could expend an additional ~350 kcal per day. In particular, obese people are described to have an innate predisposition to sit for 2.5 h per day more than lean sedentary people, possibly due to factors that induce this, mainly environmental factors [34]. NEAT varies up to 2000 kcal per day among individuals; therefore, its modulation may play an important role in increasing body mass, as obese subjects show low NEAT values.

The increased use of motorized transportation, mechanized manufacturing, and the prevalence of labor-saving technology at home and at work, combined with high sedentary behavior, decrease NEAT [35]. Modifying many of these issues is complex, but strategies can be employed to increase NEPA. For example, adding 2.5 h of walking per day for office workers (i.e., people with high sedentary activity) brings benefits for body composition and increases total physical activity without a compensatory overall effect [36]. There are potential benefits to increasing NEPA in controlling body fat, but it is important to consider the limitations of relying solely on this approach. For example, a systematic review by Silva et al. [37] found that compensatory phenomena were observed in 15 out of 36 clinical studies, such as increases in NEAT, NEPA, or both, after diet-only, combined diet/exercise, and exercise-only interventions. The degree and duration of energy imbalance generated can affect energy conservation in response to increased NEPA. Research has shown that short-duration continuous exercise programs (<40 min) can increase NEPA as a compensatory mechanism, whereas longer protocols (180 min) can lead to a decrease in NEPA. This response is more pronounced and delayed in overweight than in normal-weight individuals [38]. However, despite concerns that an increase in NEPA can lead to compensatory increases in energy intake, no increases in food cravings have been confirmed with both protocols. Furthermore, studies by Castro et al. [39] and Romero et al. [40] show that in obese populations, there is no compensatory increase in caloric intake during physical exercise programs as long as adequate nutritional guidance is provided. Therefore, an active lifestyle intervention that involves increasing NEPA and gradually progressing to physical exercise to increase caloric expenditure is a promising approach to controlling excess fat mass.

## 5. Promotion Strategies

As mentioned, physical inactivity and sedentary behavior are of high public health concern, as there is an interaction between behavioral risk factors that increase risk. For example, the constant exposure to food advertising during television viewing contributes to some extent to the modification of diet behavior, that is, the consumption of unhealthy foods (e.g., processed foods with added sugar) in both children [41] and adults [42]. In addition, a lifestyle that reduces NEPA and, thereby, energy expenditure may promote an obesogenic environment; therefore, it was reported that increasing active transportation (e.g., cycling or walking) and reducing automobile use may reduce the prevalence of obesity and NCDs [43]. In this sense, each government’s policies should encourage the use of active means of transportation in addition to incorporating measures to reduce the use of automobiles. For example, a study by Courtemanche [44] evaluated the relationship between fuel prices, body mass, and obesity rates. Higher gasoline prices were associated with a reduction in restaurant use and an increase in walking, which, in turn, was related to a decrease in body mass. Similarly, the association between fuel prices and physical activity levels in young adults (18–30 years old) was reported. For example, a $0.25 increase in fuel price is associated with increased energy expenditure (1.3 METs), which is equivalent to an increase of three min of walking per week [45]. 

Strategies to promote NEPA in the general population seek to increase physical activity levels and reduce sedentary behavior because drastically reducing NEPA increases the risk of coronary, muscular, nervous system, reproductive, digestive, immune, bone, and endocrine diseases [46,47] (Figure 2).

One of the barriers to physical activity is screen time (TV, PC, smartphone), and promoting NEPA is one strategy to reduce it. To achieve this, individuals can modify the configuration of their electronic devices by adjusting the time of use for each application or setting an on/off schedule to generate a pause in their usage period. Blocking smartphone notifications and uninstalling unnecessary applications (e.g., games and social networks) are also helpful strategies. However, for children and adolescents, positive engagement, guidance, and parental influence are key factors in reducing media-related risks [48,49]. In cases where it is not feasible to remove social networking apps for work or academic reasons, people can remove groups that encourage polarization, misinformation, outrage, and distraction [50]. Another useful strategy is enabling the grayscale display mode in Android and iOS operating system devices. This makes browsing social networks less attractive and reduces the positive reinforcement generated by the color ranges used in video games and social networks, thereby reducing screen time [51]. Additional strategies for the general population include: establishing specific schedules without any technology; creating shared spaces at home for recharging electronic devices; avoiding the use of electronic devices in bedrooms; using traditional alarm clocks to restrict smartphone use early in the morning and before going to bed; scheduling one day a week without smartphone use; avoiding smartphone use during meal times; taking a walk every day in outdoor spaces without carrying mobile devices or restricting their use in these leisure spaces [52,53].

On the other hand, the progression of time and intensity of NEPA, such as active transportation (e.g., walking), will not only identify health risk but also encourage physical activity in subjects with low adherence to physical conditioning programs [54]. For example, increasing physical activity at the expense of NEPA has clinically relevant effects for healthy (e.g., untrained) individuals and NCDs, even, in some settings, without reaching the minimum recommended physical activity [55]. Because of this, and as supported by the accumulated evidence on the multiple benefits of physical activity, not only in terms of health but also in social and economic terms, “The Global Action Plan on Physical Activity 2018–2030: More Active People for a Healthier World” [56] urges countries to implement, based on normative solutions, actions to promote physical activity and reduce sedentary lifestyles (Table 1).

It is well-established that green infrastructure in urban environments is an important element for improving health, as walking in green areas increases NEPA. Governmental entities and the construction industry should collaborate to design and develop healthy urban infrastructures that promote NEPA and improve health, not just outside residences (e.g., pedestrian walkways, parks, sports and recreational areas, and green areas). They should also consider interior design principles that promote NEPA [57]. Local public administrations can also take other measures such as closing streets for vehicular transport on certain days of the week or permanently, creating outdoor lanes for the exclusive use of cyclists and pedestrians, and relocating parking lots located within urban parks. Building a road network for the exclusive use of active transportation within cities has also been repeatedly emphasized [58,59]. 

Lastly, cities that have these spaces and projects should promote their use through different local media and encourage their implementation in daily routines. The development of salutogenic spaces within homes, workplaces, and educational facilities and their impact on health, NEPA, and NEAT should be highlighted. Designers and manufacturers are becoming more aware of the importance of creating suitable environments that promote the free mobility of older adults with functional limitations and people with disabilities or excess adiposity [60]. However, given the low implementation of spaces within homes in developing countries, the use of devices and sensors that monitor the behavior and track daily activity is recommended. Various apps can be used on cell phones to promote NEPA and facilitate the progression to regular physical exercise. Evidence suggests that individuals of different ages and sexes respond positively to using these apps as a persuasive element, facilitating a synergistic and sequential process of positive behavioral change [61]. This topic is discussed in the next section.

## 6. Technological Monitoring of Non-Exercise Physical Activity

Although the principle of NEPA does not obey a program in the strict sense, its evaluation is necessary to control the physical activity performed during a period of time, estimate caloric expenditure, set goals, and motivate people to increase NEPA. This is highly relevant for the sedentary population and for those with insufficient physical activity. For these types of populations, physical activity monitoring is often done using self-reporting methods (e.g., diaries, questionnaires, and surveys), which are very inexpensive, but their main error is recall bias [62]; therefore, the information reported by individuals may present certain impressions. For this reason, new wearable device technologies, including pedometers, accelerometers, heart rate monitors, and mobile devices, can facilitate the programming and control of physical activity [63]. Receiving information on various parameters (e.g., distances and steps taken) from these wearable devices is motivating for the user, which could be useful in facilitating the adherence to physical activity practice [64]. Moreover, this can motivate behavioral change, particularly due to the opportunities that arise to provide instant feedback [65].

Several types of wearable devices are available on the market to measure physical activity, such as accelerometers that measure the magnitude of acceleration and derived variables [66], and report the information as “counts” per unit of time [67]. Accelerometers can measure acceleration in one (unixial), two (biaxial), or three planes (triaxial) and are generally placed as close as possible to the body mass center or at the hip at the level of the mid-axillary line [67]. These devices provide an objective measure of physical activity that does not rely on self-reporting; moreover, they are superior to other wearable devices (i.e., pedometers) because they measure physical activity intensity and frequency [68]. It should be noted that movement acceleration is directly proportional to muscle strength and is related to energy expenditure [69]. Recent advances in accelerometer-based wearable devices make them a valid technology for the automatic, continuous, long-term measurement of physical activity and sedentary behavior [66,70,71]. The Michigan Predictive Activity & Clinical Trajectories in Health (MIPACT) study is a good example of how data collected from wearable devices can inform clinical trial design, interpretation in clinical practice, and health-care interventions [72]. Wearable devices represent an important element that offers new opportunities for conducting decentralized clinical trials [73], which have been shown to be effective and feasible regarding recruitment, data collection from various electronic devices, and participant engagement to monitor physiological and behaviometric (physical activity and sleep patterns) parameters [74]. For practical purposes, Table 2 summarizes some characteristics of commercially available accelerometer-based wearable devices, many of which are used in research for physical activity quantification. 

Pedometers are another type of device usually used to measure physical activity. They detect vertical accelerations of the hip by activating a lever arm to move vertically and a ratchet to rotate. They are widely used to measure the number of steps taken by a person (e.g., steps per day). This is an important consideration, given that walking is a preferred physical activity during leisure time as well as being a form of transportation, and it is recommended for most of the population, including older adults and people with NCDs. 

Pedometer use was shown to be associated with increased physical activity, decreased body mass, and lower blood pressure [76]. Furthermore, the promotion of physical activity through walking can be promoted using these devices so that monitoring the number of steps per day allows the recommended levels of physical activity to be achieved. In this regard, recommendations were made on how many steps per day are sufficient to obtain health benefits in the healthy adult population [77] (Table 3).

Many recommendations on “steps per day” to increase physical activity have as criteria the 10,000 steps per day documented by Hultquist et al. [78] in physically inactive women, who walked more when advised to achieve 10,000 steps per day compared to those instructed to walk 30 min per day. Nevertheless, several clarifications should be made: first, the initial level of physical activity, physical condition, and state of health of the individuals should be considered, given that for some, this will be an attainable goal without major problems, but for others, it could represent a difficulty (e.g., people with very poor physical condition, musculoskeletal limitations, or some pathology that limits walking). Therefore, this should be done progressively. On the other hand, the generic recommendation of 10,000 steps per day only focuses on the number of steps taken and not on the intensity of physical activity. Given this, then, we must emphasize that the recommendations to increase the number of steps per day represent one more way to promote and increase the amount of physical activity, and in this way, it should be a complementary way within the strategies to improve health and prevent or control excess body fat.

Of the two types of wearable devices mentioned previously, pedometers may be the most feasible, due to their practicality and cost-effectiveness, to determine NEPA in people’s daily lives [62]. However, the pedometer, compared to the accelerometer, cannot provide any temporal information about NEPA patterns because it does not store data [66]. Additionally, the “shuffling” gait pattern, which may be present in some older adults and in the obese population, may also contribute to pedometer errors in detecting actual steps taken; however, the use of pedometers and even smartphones that have sensors (e.g., accelerometers and gyroscopes) is recommended in the apparently healthy and obese adult population [75]. People have a greater preference for user-friendly smartphone apps because they allow automatic tracking, generate greater motivation, offer individualized feedback, and facilitate the establishment of goals, such as steps taken, distance traveled, calories expended, time of activity performed, and comparison with other users [79,80]. Among the various available apps, SMART MOVE, POWeR and POWeR Tracker, SMART MOVE GPS, short messaging service (SMS), My Meal Mate, Mobile Pounds Off Digitally (Mobile POD), Endomondo, Speedo Fit, Nike + Fuel, Pedometer++, Runkeeper, and Fitbit are some of the most popular platforms that require external sensors and personal computer use [81]. For a comprehensive review of wearable devices research trends, readers are encouraged to refer to Kageyama et al. (2022) [82].

For children or the elderly population with permanent pathologies or disabilities, the use of “smart clothing” can be a potential alternative. This technology is discreet and based on hetero-core fiber optic sensors, textile electrodes, and inertial measurement units that transmit and process information on mobile devices such as tablets or smartphones [83]. These devices monitor NEPA and other important physiological vital signs, such as heart rate, blood pressure, respiratory rate, body temperature, arterial oxygen saturation, and blood glucose, among others. Additionally, it allows for permanent biomechanical monitoring of the person through location algorithms, sensors such as gyroscopes, magnetometers, and barometric pressure sensors, among other systems, for preventing or developing a fall, body imbalance, hours of rest, and other activities of daily life. Smart shirts from Heddoko^TM^, Hexoskin, Cityzen Sciences, Ralph Lauren Polo, and Athos are good examples of this technology [84].

## 7. Conclusions

There is consistent evidence that shows the benefits of physical exercise on health and, specifically, their effects on the reduction of fat mass of the overweight and obese populations. However, the recovery from body mass loss, after a physical exercise program, constitutes an important challenge in the intervention of excess weight and obesity; additionally, strategies are required at the public health level to promote regular activity practice. In the general population, NEPA should be increased as a strategy for preventing and controlling excess fat mass by promoting the use of active transportation (e.g., walking and cycling) to increase NEAT and, thereby, total daily energy expenditure. Likewise, sedentary activities should be reduced by modifying or replacing them with activities that involve NEPA (e.g., family walks, traditional games, musical activities, and creating active environments and systems, among others). The evaluation and control of NEPA can be done through wearable devices such as smartphones, which allow tracking activity and, in this way, establishing goals regarding the fulfillment of “steps per day” and progression to physical exercise according to the individual’s ability or limitations. In line with this, improvements to NEPA should consider aspects of motivation, tastes for types of activity, and the ease with which it can be carried out (e.g., schedules, working hours, facilities, and urban designs), with which the aim is to achieve adherence and changes in the subject’s habits and lifestyles. Thus, NEPA constitutes a sustainable long-term strategy for preventing and controlling obesity as a complement to physical exercise (which should include strength training) and nutrition programs.

Although promoting NEPA is an important strategy for preventing and controlling obesity, addressing the systemic factors contributing to sedentary behavior and physical inactivity is also critical. Creating active environments, such as safe and accessible parks and bike lanes, and promoting policies that incentivize physical activity, such as workplace wellness programs and school physical education requirements, are examples of how this can be done. Additionally, efforts should be made to address structural inequities and barriers that prevent certain populations, such as low-income and minority communities, from participating in physical activity. We can work toward a more comprehensive and effective approach to reducing excess weight and obesity and promoting health for all by addressing these broader factors and promoting NEPA at the individual level.

## Figures and Tables

**Figure 1 jfmk-08-00044-f001:**
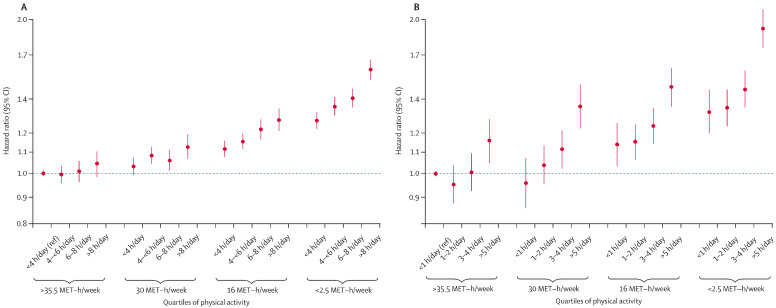
Associations of sitting time, television viewing time, and physical activity with all-cause mortality. (**A**) Analysis of sitting time (*n* = 1,005,791) and (**B**) television viewing time (*n* = 465,450) as well as their relationship to physical activity levels and mortality risk [29].

**Figure 2 jfmk-08-00044-f002:**
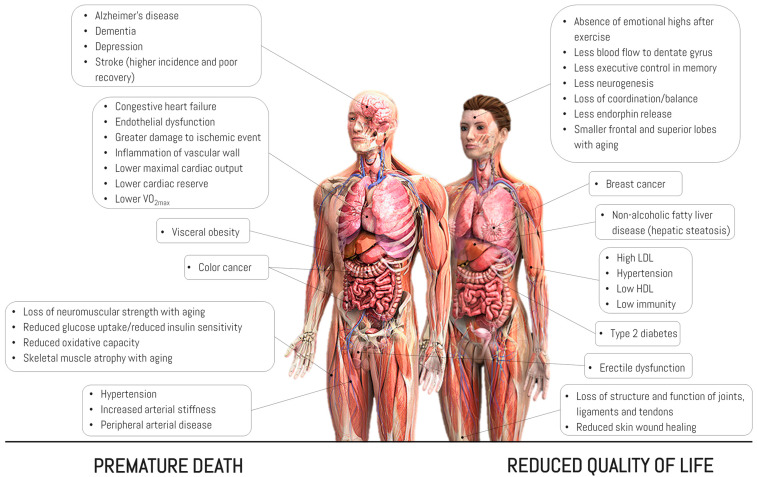
Effects when moving from high to low levels of physical activity, including significant reductions in NEPA. Source: designed by the authors (D.A.B.) based on published materials [46].

**Table 1 jfmk-08-00044-t001:** Public health objectives and actions to promote physical activity and reduce sedentary lifestyles.

Target	Components	Descriptions/Actions
Creating physically active societies	Social norms and attitudes	To achieve a paradigm shift throughout society by improving the knowledge, understanding, and appreciation of the multiple benefits of regular physical activity according to ability and at all ages.
Creating active environments	Spaces and places	To create and maintain environments that promote and safeguard the rights of all people, of all ages, to enjoy equitable access to safe places and spaces in their cities and communities where they can engage in regular physical activity according to their abilities.
Promoting active populations	Programs and opportunities	To create and promote access to opportunities and programs, in multiple settings, to help people of all ages and abilities participate regularly in physical activity, either alone or with their families and communities.
Creating active systems	Governance and policy enablers	To enable elements of governance and policy to build and strengthen leadership, governance, multisectoral partnerships, workforce capacities, advocacy, and information systems across sectors to achieve excellence in resource mobilization and the implementation of coordinated international, national, and subnational actions to increase physical activity and reduce sedentary lifestyles.

Adapted from WHO [56].

**Table 2 jfmk-08-00044-t002:** Characteristics of some accelerometer-based portable devices.

Device	Sensor	Sensor Location	Advantages/Disadvantages	Variables
SenseWear, BodyMedia Inc. (CAM)	2-axis accelerometer	Upper arm	High concordance to assess PA and EE. Requires specific software to interpret data. Low sensitivity in subjects with functional mobility limitations.	Acceleration and EE (METs).
CT1/RT3, StayHealthy Inc. (CAM/ RGAM)	3-axis accelerometer	Wrist or hip (RT3)	Detailed information on activities of daily living. May present difficulties in manipulation (switching on/off) in older individuals or persons with disabilities or mobility difficulties.	Activities of daily living (NEPA) and displacements (as vector magnitude units).
AMP331, Dynastream Innovations Inc. (CAM)	2-axis accelerometer	Ankle	Due to its dimensions, it is comfortable to use for several consecutive days. Useful to control the intensity of physical exercise.	Activities of daily living (NEPA), vertical and horizontal accelerations, number of steps, frequency and stride speed, and EE.
wGT3X-BT, Actigraph LLC (RGAM)	3-axis accelerometer, time-of-use sensor, ambient light sensor	Wrist or hip	High concordance to assess PA and EE. Small device and easy location. Requires specific software to interpret data. Overestimates EE when using motor vehicle transportation.	Acceleration (as vector magnitude units) and EE.
StepWatch, Orthocare Innovations (CAM)	2-axis accelerometer	Ankle	Portable device designed to track and monitor physical activity. Useful for persons with lower-limb disabilities or mobility difficulties. High precision and measures PA on different surfaces such as soil, grass, carpets, etc. High cost compared to other devices. It needs to be calibrated to ensure accurate results. Some people find it uncomfortable to wear the device on their ankle or wrist all day.	Activities of daily living, duration and pause time between them (walking, jogging, running or sprinting, and sitting or standing), EE, steps per minute, moderate-to-vigorous PA, and total PA.
activPAL, PAL Technologies Ltd. (CAM)	Accelerometer	Thigh	Useful for different populations (children, older adults, and patients with chronic diseases) and more user-friendly. Expensive device. Must be placed on upper thigh, which is uncomfortable for some users. Not water-resistant.	Activities of daily living (NEPA) and displacements (body inclinations in degrees). Light and moderate-to- vigorous PA.
IDEEA, MiniSun (CAM)	2-axis accelerometer	Chest, thigh, or ankle	Uses several sensors on different parts of the body at the same time (foot, ankle, thigh, and chest). Requires constant adjustment of the sensors to avoid loss of information. Underestimates EE in continuous static arm activities such as cycling or arm exercises and slow walking. Slightly overestimates EE in other NEPA activities. Not water-resistant.	Activities of daily living (NEPA), HR, EE, and acceleration.
Inspire, Fitbit Inc. (CAM)	3-Axis Accelerometer	Wrist or hip	Compatible with a variety of applications (apps) and fitness platforms. Low cost. Not very accurate in PA measurement and does not measure HR continuously, limiting the measurement of moderate-to-vigorous PA. Low battery life.	Activities of daily living, duration and pause time between them (walking, jogging, running or sprinting, and sitting or standing), and EE.
VivoFit 4, Garming Ltd. (CAM)	Accelerometer	Wrist	Compatible with the Garmin Connect app and is water-resistant. No GPS and no HR.	Activities of daily living (NEPA), sleep quality, EE, and total PA.
Vivomove HR, Garming Ltd. (CAM)	Accelerometer, barometer, photoplethysmography, ambient light sensor	Wrist	Compatible with the Garmin Connect app, long battery life, and water-resistant.	Activities of daily living (NEPA), EE and total PA, and sleep monitoring.
Mi Band 3, Xiami Corp. (CAM)	3-axis accelerometer, photoplethysmography	Wrist	Compatible with the Mi Fit app, water-resistant, and affordable price. No GPS and no HR.	Activities of daily living (NEPA), EE, and total PA.
Pulse HR, Withings (CAM)	3-axis accelerometer, photoplethysmography, ambient light sensor	Wrist	Multisport tracking, connected GPS, and an OLED screen that displays full smartphone notifications for calls, texts, events, and all of your favorite apps.	HR, training zones, and sleep quality.
Steel HR, Withings (CAM)	3-axis accelerometer, day and night motion sensor	Wrist	Compatible with the Health Mate app, water-resistant, and call and message notification.	Activities of daily living (NEPA), EE and total PA, sleep monitoring, EE, and HR.
TriTrac-R3D, Madison, WI, USA	3-Axis Accelerometer	Hip	Good concordance to assess PA and EE during physical exercise. Very high correlation when evaluating HR in children. Does not show good accuracy in sedentary individuals. Requires software to estimate total EE (kJ/min).	Activities of daily living (NEPA), EE, and acceleration.

CAM: consumer activity monitor; EE: energy expenditure; GPS: Global Positioning System; HR: heart rate; METs: metabolic equivalents; PA: physical activity; RAM: research-grade activity monitor. Based on [71,75].

**Table 3 jfmk-08-00044-t003:** Classification of physical activity based on the number of steps per day.

Steps per Day	Physical Activity Level
<5000	Insufficiently active
5000–7499	Somewhat active
7500–9999	Moderately active
>10,000	Active
>12,500	Highly active

Adapted from Tudor-Locke and Bassett [77].

## Data Availability

Not applicable.

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
