# Peer review of "Insights into Non-Exercise Physical Activity on Control of Body Mass: A Review with Practical Recommendations"

_jfmk, 2023, doi:10.3390/jfmk8020044_

Round 1
Reviewer 1 Report
I find this topic very popular nowadays, therefore i am not sure about the novelty of this review. Nontheless I find this work, in general, well written. Although I have some concerns, mainly to references.
line 73-74 - The authors mentioned they based on Dixon-Woods and Popay works, but they are both wrongly cited. Number 17 and 18 do not correspond properly to sources listed in reference section.
line 98 - reference number [17] relates to Piggin's work and Popay's work.
Reference numbers have to be revised and correct.
References:
51 out of 75 references are up-to-date (not older than 10 years), there are also some very new, from 2022 which proves that the topic is currently popular and the authors have prepared the article on the basis of the most up-to-date data.
However, the list of links contains numerous errors, it lacks access date information, and some links are incorrect (e.g. no. 18)
When preparing a review article, one of the most important things is correct citation, because this is what ensures other authors will find the original publication and, in example to check the study design.
Author Response
Dear reviewer,
Thanks for your review report.
Attached you will find the response to your comments.
Sincerely,
The Authors

Reviewer 2 Report
Dear Authors,
I think it's this review that summarizes much literature.
However, I would like you to write a description in the “introduction” that emphasizes the new perspectives for conventional reviews.
Author Response
Dear Reviewer,
Thanks for your review report.
Below you will find the response to your comments.
----------------
I think it's this review that summarizes much literature.
Response: We would like to thank the reviewer for the comments.
However, I would like you to write a description in the “introduction” that emphasizes the new perspectives for conventional reviews.
Response: Thanks for your suggestion. We have written some statements regarding the limitations of non-systematized narrative reviews. Therefore, several amendments have been performed but were placed in the first part of the “Methods” section for readability. All changes have been clearly highlighted in red (tracking changes) so that they can be easily visible to the editor and reviewers.
-------------
Sincerely,
The Authors
Round 2
Reviewer 1 Report
I thank the Authors and I am glad that the bibliography has been improved, both the references in the text and the bibliography itself. However, references 16, 20 and 22 still lack access date information. After this correction, I recommend the work for publication.
Author Response
Dear Reviewer,
Thanks for notifying this typo.
We have addressed it.
Sincerely,
The Authors